# Antibody Generation and Rapid Immunochromatography Using Time-Resolved Fluorescence Microspheres for Propiconazole: Fungicide Abused as Growth Regulator in Vegetable

**DOI:** 10.3390/foods11030324

**Published:** 2022-01-24

**Authors:** Bo Chen, Xing Shen, Zhaodong Li, Jin Wang, Xiangmei Li, Zhenlin Xu, Yudong Shen, Yi Lei, Xinan Huang, Xu Wang, Hongtao Lei

**Affiliations:** 1Guangdong Province Key Laboratory of Food Quality and Safety, College of Food Science, South China Agricultural University, Guangzhou 510642, China; 13971356810@163.com (B.C.); shenxing325@163.com (X.S.); wangjin940810@stu.scau.edu.cn (J.W.); lixiangmei12@163.com (X.L.); xzlin@scau.edu.cn (Z.X.); shenyudong@scau.edu.cn (Y.S.); 2College of Materials and Energy, South China Agricultural University, Guangzhou 510642, China; scaulizhaodong@scau.edu.cn; 3Guangdong Institute of Food Inspection, Zengcha Road, Guangzhou 510435, China; Leiy04@foxmail.com; 4Tropical Medicine Institute and South China Chinese Medicine Collaborative Innovation Center, Guangzhou University of Chinese Medicine, Guangzhou 510405, China; xahuang@chinmednetworks.org; 5Institute of Quality Standard and Monitoring Technology for Agro-Products of Guangdong Academy of Agricultural Sciences, Guangzhou 510405, China; wangxuguangzhou@126.com

**Keywords:** propiconazole, hapten, antibody, time-resolved fluorescence, lateral flow immunochromatographic assay

## Abstract

Propiconazole (PCZ) is a fungicide popularly used to prevent and control wheat and rice bakanae disease, etc. However, it was recently found to be illegally employed as a plant regulator to induce thick stems and dark green leaves of *Brassica campestris*, a famous vegetable in Guangdong, South China. Due to a lack of available recognition molecules to the target analyte, it is still a big challenge to establish a rapid surveillance screening method. In this study, a novel chiral hapten was rationally designed, and an artificial immunogen was then prepared for the generation of a specific antibody against propiconazole for the first time. Using the obtained antibody, a highly sensitive time-resolved fluorescence microspheres lateral flow immunochromatographic assay (TRFMs-LFIA) was established with a visual limit of detection of 100 ng/mL and a quantitative limit of detection of 1.92 ng/mL for propiconazole. TRFMs-LFIA also exhibited good recoveries ranging from 78.6% to 110.7% with coefficients of variation below 16%. The analysis of blind real-life samples showed a good agreement with results obtained using HPLC-MS/MS. Therefore, the proposed method could be used as an ideal screening surveillance tool for the detection of propiconazole in vegetables.

## 1. Introduction

Propiconazole (PCZ) is a systemic triazole fungicide that can prevent most fungal diseases in banana, wheat and rice [1]. It is favored by farmers for its broad-spectrum sterilization and long duration. The crops registered for using of propiconazole in China are banana, wheat and rice [2]. In 2018, the annual use of propiconazole in China reached 2087.42 tons, which exceeded other countries [3]. In the hazard classification, propiconazole is classified as reproductive toxicity 1B category and belongs to endocrine disrupting substances [4]. Related studies have confirmed that propiconazole can result in genetic toxicity, liver toxicity and growth toxicity [5,6,7]. The U.S. Environmental Protection Agency has included propiconazole in the list of possible human carcinogens [8].

Most Cantonese prefer brassica campestris with thick stems and a dark green color, which may be the most favored vegetable by local people in Guangdong due to its rich nutrition and taste [9]. However, in order to cater to the preferences of Cantonese, propiconazole was sometimes illegally used as a plant growth regulator to obtain a good appearance of brassica campestris [10]. During the first three quarters of 2016, an average of 0.011~0.07 mg/kg propiconazole was detected brassica campestris samples from 11 farmer markets and supermarkets in Guangzhou (capital city of Guangdong province, China) [11]. Based on 84 brassica campestris samples from nine farmer markets in Guangzhou, the average residue of propiconazole was 0.06 mg/kg, and the average positive detection rate was as high as 90% [12]. All these indicate propiconazole residues are commonly found in brassica campestris in Guangzhou markets.

Japan stipulates that the maximum residue of propiconazole in brassica and leafy vegetables is 0.05 mg/kg [13]. The EU cancelled the registration of propiconazole in 2019 due to a lack of data on the toxicity evaluation of propiconazole metabolites, and the existing data could not complete the risk assessment related to consumer dietary intake [14]. According to the China food safety standard “Maximum Residue Limits of Pesticides in Foods” (GB 2763-2019), some vegetables and fruits have been set a maximum residue limit for propiconazole residues, e.g., typhalatifolia (0.05 mg/kg) and banana (1 mg/kg). However, there is no residue limit set for propiconazole in many other vegetables.

Methods for detecting propiconazole include gas chromatography (GC) [15,16], high-performance liquid chromatography (HPLC) [17,18], gas chromatography-tandem mass spectrometry (GC-MS/MS) [19,20], and liquid chromatography-tandem mass spectrometry (LC-MS/MS) [21,22]. These methods are widely used due to their advantages in sensitivity and accuracy [23]. However, they are complex, require professionals to operate, and are time- and cost-sensitive [24]. The immunoassay has become the most important part of the rapid detection field because of its time-saving, high sensitivity, and simple operation [24]. Until now, there are limited reported immunoassays for propiconazole detection, mainly colloidal gold immunochromatographic assay (CG-ICA) [25] and indirect competitive enzyme-linked immunosorbent assay (ic ELISA) [26]. Nevertheless, the ELISA method requires laboratory conditions and tedious washing steps. In order to establish a highly sensitive as well as convenient propiconazole immunoassay, fluorescent microspheres-based immunochromatography has great potential.

In this study, a portable and rapid immunochromatographic assay using time-resolved fluorescent microspheres as a tracer was established for the detection of propiconazole. The hapten used in the immunochromatographic assay was rationally designed using a similar chiral carbon containing structure to that of propiconazole, and a polyclonal antibody that specifically recognizes propiconazole was obtained. Based on this antibody, a time-resolved fluorescence microspheres lateral flow immunochromatographic assay (TRFMs-LFIA) was developed, optimized, and evaluated for its sensitivity, specificity, and recovery, and then applied for the analysis of blind vegetable samples.

## 2. Materials and Methods

### 2.1. Reagents and Solutions

Propiconazole (PCZ), difenoconazole, diniconazole, hexaconazole, tebuconazole, epoxiconazole, myclobutanil, paclobutrazol, flusilazole, cyproconazole, triadimenol, bitertanol Standard, triadimefon, N,N-Dimethylformamide (DMF), N-hydroxysuccinimide (NHS), 1-ethyl-3-(3-dimethylaminopropyl)-carbodiimide (EDC), ovalbumin (OVA), keyhole limpet hemocyanin (KLH), bovine serum albumin (BSA), graphitized carbon black (GCB), N-Propylethylenediamine (PSA), 4-Dimethylaminopyridine (DMAP), N,N-Carbonyldiimidazole (CDI), succinic anhydride anhydrous dichloromethane, and 2-(N-morpholino) ethanesulfonic acid (MES) were purchased from Sigma-Aldrich (St. Louis, MO, USA). ((2S,4S)-2-((1H-1,2,4-triazol-1-yl) methyl)-2-(2,4-dichlorophenyl)-1,3-dioxolan-4-yl) methanol (AZC) was purchased from Toronto Research Chemicals (Toronto, Canada). Time-resolved fluorescence microsphere (TRFM), europium chelate (365/610), 0.2 μm, 1% (*w/v*) solid suspension, was purchased from Bangs Laboratories Inc. (Indiana, USA). Nitrocellulose filter membrane (Sartorius, UniSart CN95) were obtained from Sartorius Stedim Biotech GmbH (Goettingen, Germany). Sample pad (GF-2), absorbent pad (CH37 K), adhesive backing pad (SMA31-40), and goat anti-rabbit-immunoglobulin G (IgG) were purchased from Shanghai Liangxin Co. Ltd. (Shanghai, China). Microtiter plates were obtained from the Guangzhou JET BIOFIL Co. (Guangzhou, China). Ultra-pure water was produced using a Unique R-10 water purification system (Unique R-10, Bedford, MA, USA). Chloroauric acid, trisodium citrate, polyvinyl pyrrolidone (PVP), and other chemical substances were purchased from Sinopharm Chemical Reagent Co., Ltd. (Shanghai, China).

New Zealand White Rabbit were purchased from the Guangdong Medical Experimental Animal Centre and raised at the Animal Experiment Centre of South China Agriculture University (Animal Experiment Ethical Approval Number: 2020009, Appendix A). All required licenses were secured prior to commencement of the animal experiments.

### 2.2. Apparatus

The BioDot-XYZ 3060 Dispensing Platform was supplied by BioDot, Inc, (Irvine, CA, USA). The programmable strip cutter ZQ-2000 was purchased from Shanghai kinbio Tech. Co., Ltd. (Shanghai, China). The Lynx 4000 centrifuge was supplied by Thermo Fisher Scientific GmbH (Berlin, Germany). An FIC-Q1 multifunctional fluorescence reader was purchased from Fenghang technology Co., Ltd. (Hangzhou, China). The Nano Drop 2000C ultra-violet spectrophotometer was supplied by Thermo Scientific (Waltham, MA, USA). The UV spectrometer was purchased from Qiangyun Co. (Shanghai, China). The Zetasizer Nano ZS90 used for measurements of size and charge of nanoparticles was supplied by Malvern Panalytical (Malvern, UK). Agilent 1290-6470 Liquid Mass Spectrometry (USA AB SCIEX).

### 2.3. Synthesis of Hapten AZC-HS

The scheme for the hapten synthesis is shown in Figure 1A. Briefly, AZC (0.5 g, 1.5 mmol) was dissolved in 10 mL of anhydrous dichloromethane in a flask. After adding succinic anhydride (0.3 g, 3 mmol) and DMAP (0.018 g, 0.15 mmol), the mixture was stirred at room temperature for 12 h. Then, the mixture was evaporated to dryness before introducing water, extracted with ethyl acetate, and the organic phase was separated, dried over anhydrous Na_2_SO_4_, and finally concentrated to afford the hapten, 4-(((2S,4R)-2-((1H-1,2,4-triazol-1-yl) methyl)-2-(2,4-dichlorophenyl)-1,3-dioxolan-4-yl) methoxy)-4-oxobutanoic acid (AZC-HS). ESI-MS analysis (negative): m/z 428.8 [M − H]^−^; ^1^H NMR (600 MHz, DMSO-d6) δ 12.28–12.23 (m, 1H), 8.40 (d, J = 5.2 Hz, 2H), 7.84 (d, J = 5.1 Hz, 2H), 7.66 (d, J = 5.3 Hz, 2H), 7.46 (t, J = 6.7 Hz, 2H), 7.41 (q, J = 10.2, 7.0 Hz, 2H), 4.84–4.74 (m, 4H), 4.23 (q, J = 5.6 Hz, 2H), 3.98 (dd, J = 10.8, 5.6 Hz, 2H), 3.85 (ddd, J = 26.8, 12.4, 5.9 Hz, 5H), 3.65 (dt, J = 11.7, 5.7 Hz, 2H), 2.55 (t, J = 6.0 Hz, 4H), 2.50 (q, J = 6.6, 6.2 Hz, 6H), 2.42 (d, J = 5.3 Hz, 1H), 1.23 (d, J = 5.4 Hz, 1H).

### 2.4. Preparation of Immunogen and Coating Antigen

The immunogen was synthesized using an active ester method [27] with slight modifications. Briefly, 20 mg of hapten AZC-HS was dissolved in 3 mL of DMF. Then, 12.84 mg of EDC and 10 mg of NHS were added and kept stirring at room temperature for 5 h. The mixture was added dropwise to a reaction flask containing KLH (10 mg/mL, 1 mL), and stirred at 4 °C for 12 h. The reaction mixture was purified by dialysis against PBS (0.01 M, pH 7.4) for 3 days to remove the non-reacted reactants. The dialyzed product was the immunogen (AZC-HS-KLH). Full wavelength ultraviolet-visible (UV-Vis) spectroscopy scan was used to confirm the conjugation of the immunogen (Appendix A), which was finally stored at −20 °C until use.

Coating antigens were prepared using a CDI method [28], OVA was used as the carrier (Figure 1B). Briefly, 110 mg of AZC was dissolved in 3 mL of DMF. Then, 15 mg of CDI and 5 mg of DMAP were added with stirring at room temperature for 24 h. After that, the mixture was added to dropwise to a reaction flask containing OVA (10 mg/mL, 1 mL), and was stirred at 4 °C for 12 h. The reactive mixture was also dialyzed against PBS (0.01 M, pH 7.4) at 4 °C for 3 days. The dialyzed product was the coating antigen (AZC-CDI-OVA) (Appendix A) and confirmed by UV-Vis scan as above before being stored at −20 °C until use.

**AZC**, ((2S,4S)-2-((1H-1,2,4-triazol-1-yl) methyl)-2-(2,4-dichlorophenyl)-1,3-dioxolan-4-yl) methanol.

**AZC-HS**, 4-(((2S,4R)-2-((1H-1,2,4-triazol-1-yl) methyl)-2-(2,4-dichlorophenyl)-1,3-dioxolan-4-yl) methoxy)-4-oxobutanoic acid.

**OVA**, Albumin from chicken egg white. **CDI**, N,N′-Carbonyldiimidazole.

### 2.5. Antibody Generation

As shown in Figure 2A, New Zealand white rabbits (6~7 weeks age) were immunized with the immunogen by subcutaneous injection to the neck and back, as described previously [29]. The immune effect was verified by detecting the titer and inhibition of the rabbit serum by the competitive indirect enzyme linked immunosorbent assay [30].

The caprylic acid-ammonium sulfate precipitation method [31] was used to purify the antibody. The purified product was stored at −20 °C. Sodium dodecyl sulfate-polyacrylamide gel electrophoresis was used to check the purity of the purified antibody. The performance of antibody was preliminarily evaluated based on the 50% inhibition concentration (IC_50_) by competitive indirect enzyme-linked immunosorbent assay [32].

### 2.6. Molecular Surface Electrostatic Potential Simulation

Energy-minimized three-dimensional (3D) structure and surface electrostatic potential iso-surfaces of hapten AZC-HS and 13 triazoles compounds were modelled using the Sybyl-X 2.0 program package (Tripos Inc, St. Louis, MO, USA).

### 2.7. Preparation of TRFM Labeled Antibody

The carboxyl group on the surface of TRFM can be linked to the amino group in the antibody by the active ester method [33]. Hence, 10 μL of TRFM was mixed with 1 mL of the MES solution (50 mM, pH 5.5). Then, 15 μL of freshly prepared EDC solution (0.5 mg/mL) and 20 μL of freshly prepared NHS solution (0.5 mg/mL) were added to the above solution. The mixture was vortexed for 10 s in order to fully disperse in solution. After activation for 15 min, the solution was centrifuged at 14,000 rpm at 4 °C for 15 min. After carefully discarding the supernatant, the white precipitate was resuspended with a boric acid buffer (BB, 20 mM, pH 8.0), and the anti-PCZ antibody (2 μL, 17.0 mg/mL) dissolved in 198 μL of BB (2 mM, pH 8.0) was added and mixed thoroughly. After incubation for another 45 min, 20 μL of 20% (*w/v*) BSA was added dropwise for 60 min of blocking reaction. The mixture was then centrifuged at 14,000 rpm at 4 °C for 15 min, and the supernatant was discarded to remove any unbound antibody and BSA. The white precipitate, which was the target TRFM-labeled antibody (TRFM-Ab) immunoconjugates, was redissolved in 200 μL phosphate buffer (PB, 20 mM, pH 7.4) containing 0.75% (*v/v*) Tween-20, 0.05% (*w/v*) NaCl, 0.5% (*w/v*) BSA, 0.3% (*w/v*) PVP, and 0.03% (*w/v*) procline-300. The resuspension was stored at 4 °C for the further use.

### 2.8. Fabrication of the Lateral Flow Strip

As shown in Figure 2B, the test strip mainly consists of four parts, a sample pad, a nitrocellulose membrane, an absorbent pad, and an adhesive backing pad. First, the coating antigen was diluted to 0.2 mg/mL with carbonic acid buffer and the goat anti-rabbit IgG antibody was diluted to 0.03 mg/mL with PB (20 mM, pH 7.4). Second, they were dispensed on the nitrocellulose membrane as the test (T) line and control (C) line, with 0.8 μL/cm of spray volume, respectively. The distance between the T and C line was 8 mm, then the nitrocellulose membrane was dried at 37 °C for 12 h. Third, the nitrocellulose membrane, sample pad, and absorbent pad were adhered to the adhesive backing pad as shown in Figure 2B. Finally, the assembled backing pad was cut into strips with widths of 3.05 mm and placed in a ziplock bag with silica particles as a desiccant.

### 2.9. Sample Preparation

Blank lettuce and romaine lettuce purchased from a local supermarket were verified as PCZ-free by HPLC analysis.

The standard addition process is to add the standard analyte of the corresponding concentration to 2 g of homogeneous vegetables, vortex, and mix for 1 min.

Samples were extracted by mixing 2 g of homogenized vegetables with 5 mL of methanol in 10 mL centrifuge tubes. After mixing and shaking for 1 min, the extracted solutions were centrifuged at 5000 rpm for 5 min. The supernatants of 4 mL were collected in a 10 mL centrifuge tube (containing 50 mg PSA, 30 mg GCB). After mixing and shaking for 1 min, the extracted solutions were centrifuged at 5000 rpm for 5 min. The supernatants of 2 mL were collected and dried with nitrogen. The substrate was reconstituted with 2 mL of PB (200 mM, pH 7.4) containing 10% (*v/v*) methanol, before being filtered through a 0.22 μm membrane and then diluted four times with PB (200 mM, pH 7.4) containing 10% (*v/v*) methanol. The final volume of 8 mL used to for TRFMs-LFIA analysis.

### 2.10. Principle and Detection Protocol of TRFMs-LFIA

The main principle of TRFMs-LFIA is based on the indirect competitive reaction between antibody and antigen [34]. The TRFM-Ab acts as a sensitive fluorescent probe. As shown in Figure 2B, with the help of absorption capacity of the absorbent pad, the solution can flow in the direction from the sample pad to the absorbent pad. When detecting a positive sample, the free propiconazole first reacts with the TRFM-Ab conjugates and occupies the limited TRFM-Ab binding sites. Thus, fewer or no TRFM-Ab conjugates will be captured by coating antigen on the T line. The more propiconazole exists in the sample, the weaker fluorescence response displays on the T line. In accordance with this principle, the fluorescence intensity at the test line is inversely proportional to the propiconazole concentration in the sample. If there is no fluorescence on the C line, the test strip is invalid (Figure 2C).

In this study, a vertical running mode was applied to the test strip for the detection process. In our work, 120 μL of sample solution or standard analyte was added to the microwells (microtiter plates, Guangzhou JET BIOFIL Co.), followed by mixing with 8 µL of the TRFM-Ab. After incubation for 3 min at room temperature, the test strips were vertically inserted into the microwells and incubated for another 5 min. Finally, the sample pad was discarded, and the test strips were visually inspected under ultraviolet (UV) light (Figure 2C) for qualitative results. Meanwhile, the fluorescence intensity of the T and C line were quantified by a FIC-Q1 multifunctional fluorescence reader (Figure 2D) for quantitative analysis. The ratio of T/C was used to quantify the result, where T represent fluorescence intensity at the detection line [35].

### 2.11. Performance of TRFMs-LFIA

A series of concentration of propiconazole standard solutions (0, 5, 25, 50, 75, 100, 150, and 200 ng/mL) under the optimal conditions was analyzed using the established TRFMs-LFIA to achieve the visual limit of detection (vLOD), standard curves, and quantitative limit of detection (qLOD). The vLOD, judged by the naked eyes under the UV light, is considered to be as the lowest analyte concentration that can form a significantly weaker color band on the T line than that on the control strip [36]. The standard curves were obtained by measuring a series of standard with various concentrations. Each level was tested in triplicate (*n* = 3) with the strip by FIC-Q1 fluorescence reader, which were plotted based on the B/B_0_ against the propiconazole concentration, where B and B_0_ were the ratio of T/C values with and without propiconazole in the sample solutions. The qLOD were defined as the concentration that gave 80% B/B_0_ values according to the standard curves [37].

To evaluate the specificity of TRFMs-LFIA, 12 structural and functional analogs, including difenoconazole, diniconazole, hexaconazole, tebuconazole, epoxiconazole, myclobutanil, paclobutrazol, flusilazole, cyproconazole, triadimenol, bitertanol standard, and triadimefon, were tested along with propiconazole at the same concentration (100 ng/mL), and PB (0.1 M, 10% (*v/v*) methanol) was selected as the negative control. Each test was repeated in triplicates.

In order to confirm the reliability of the proposed TRFMs-LFIA, spiked and real-life samples were verified by HPLC-MS/MS analysis. The accuracy and precision of the developed test strip was evaluated by the recovery and coefficient of variation (CV), respectively [38]. At present, the residues of propiconazole in the brassica campestris seem to be common, and in order to better evaluate the residues of propiconazole in markets, two extra leafy vegetables, lettuce and romaine lettuce, were also selected as the test samples. Brassica campestris, lettuce, and romaine lettuce samples, including three levels of propiconazole standard concentration (10, 40, 80 μg/kg), were detected via test strip in triplicates. Blind samples (four samples of each vegetable) taken randomly from local markets in Guangzhou were tested in triplicates by both TRFMs-LFIA and HPLC-MS/MS. The quantitative consistency of TRFMs-LFIA and HPLC-MS/MS was verified by regression analysis.

## 3. Results and Discussion

### 3.1. Hapten Design and Antibody Evaluation

As propiconazole is too small a molecule (molar mass 342.2 g mol^−1^) to be immunogenic and to exert an immune response to an animal body, the hapten needs to be conjugated to a macromolecular protein carrier to become a complete antigen for animal immunity [39]. However, propiconazole lacks an active group for conjugation with a carrier protein. Therefore, the structurally similar analogue molecule AZC was selected as a hapten, which extended an arm similar to propiconazole feature on its side chain. There are two chiral carbons in the molecular structure of propiconazole, and it is well known that the propiconazole contains four optical isomers. The cis-isomer generally accounts for about 60% of the racemate, and the trans-isomer accounts for about 40% [40]. Herein, AZC-HS, the cis isomer (Figure 2) was chosen as the immunizing hapten for the conjugation with carrier protein to prepare the immunogen, and then immunize rabbit to produce an antibody. It was expected to recognize the propiconazole isomer molecule, since the majority of their features are same except for a steric difference and the length of the side chain in their structures (Figure 1A).

In this conjugation, the overall structure of propiconazole was fully exposed by introducing the succinic anhydride arm, which was coupled with protein (KLH) to obtain immunogen and the conjugate curves (Appendix A) showed the conjugate has the weak characteristic peaks of both the hapten at 260 nm and the carrier protein at 280 nm, preliminarily judging that the coupling is successful. Meanwhile, the hapten AZC was directly coupled to the OVA to obtain the coating antigen, and the conjugate curves (Appendix A) showed that the conjugate has the characteristic peaks of both the hapten at 320 nm and the carrier protein at 280 nm, indicating successful conjugation between the hapten and the carrier (BSA or OVA) [41], thus supporting the formation of the immunogen. This immunogen was then used for antibody production.

After the rabbit antiserum was obtained, preliminary purification and evaluation were carried out per the previously reported protocol [31]. SDS-PAGE under reducing conditions (Appendix A) shows the main protein band between 55 kDa and 180 kDa. The propiconazole antibody showed two expected protein bands around 25 kDa and 55 kDa, respectively, indicating a normal antibody feature, and consequently high antibody purity.

A competitive indirect enzyme linked immunosorbent assay was developed to preliminarily assess the usability and suitability of the obtained antibody with IC_50_, which is the concentration resulting the half decrease in optic absorbance [32] (Appendix A). The performance of the competitive indirect enzyme linked immunosorbent assay calibration curve exhibited a good property with IC_50_ values of 0.51 ng/mL, a working range (IC_20_~IC_80_) of 0.11~2.42 ng/mL, and a limit of detection (LOD) 0.06 ng/mL. According to the China food safety standard “Maximum Residue Limits of Pesticides in Foods” (GB 2763-2019), the maximum residue limits of propiconazole in other agricultural products is 0.05 mg/kg, the obtained performance of the antibody is much lower than the limit set by GB 2763-2019. This indicated that the antibody is sensitive enough to meet the limit regulation and suitable for the further investigation and assay development.

### 3.2. Characterization of TRFM Labeled Antibody

When TRFM was coupled with antibody, the coupling can sometimes cause a change in the particle size and zeta potential, which indicates a successful conjugation [42]. Here, the particle size is shown in Appendix A. When the 10 μL of TRFM and TRFM-Ab was dispersed in 1 mL of ultrapure water, an average particle size of TRFM was approximately 200 nm. After coupling, its average particle size of TRFM-Ab was about 240 nm. There is a remarkable change in particle size before and after coupling. With regard to zeta potential, the potential of TRFM-Ab increased significantly from −50 mV to −35 mV of TRFMs (Appendix A). This indicated that the antibody was successfully labeled with the TRFMs.

### 3.3. Optimization of TRFMs-LFIA

To obtain the best performance, each step of the TRFM-LFIA assay procedure was optimized. In this study, several key factors were optimized, such as the conditions for the preparation of TRFM-Ab (activation pH, antibody dilution buffer antibody amount, etc.) and the assay procedure (probe amount, ion concentration of standard diluent, methanol content in diluent, coupling pH, time of coupling reaction and blocking time, etc.).

#### 3.3.1. Activation pH

The carboxyl group on the surface of the TRFMs requires activation to couple with the amino group of the antibody [43]. Therefore, the activated pH is the key factor affecting the efficiency of activation. The activation pH was adjusted to 5.0, 5.5, 6.0, and 6.5 with MES (0.05 M) solution and other pH was adjusted to 7.0 and 7.4 with PB (0.01 M), respectively. The result showed that with the increase of pH, the fluorescence intensity of C line first increases and then decreases (Figure 3A). When the pH was at 5.5 during preparation of the TRFM labeled antibody, the fluorescence intensity of C and T lines was desired under the UV light. From the histogram analysis, the inhibition rate is the highest at the same time. Therefore, pH 5.5 was selected as the optimal activation pH.

#### 3.3.2. Antibody Dilution Buffer

It is known that a suitable dilution type is an important factor that can affect the sensitivity and specificity of antibody [44]. In this investigation, five antibody dilution buffers, including ultrapure water, 0.01 M PB (pH 7.4), 0.5% BSA, 0.01 M PB (0.5% BSA), and 0.002 M BB (pH 8.0), were used to dilute antibody. The results under UV light (Figure 3B) showed that the color rendering effect was the worst when the antibody dilution buffer contains 0.5% BSA, while the antibody dilution buffer contains a small number of ions and leads to a better color rendering effect. From the histogram analysis, the inhibition rate is the highest when the antibody dilution buffer was 0.002 M BB. Therefore, 0.002 M BB (pH 8.0) was selected as the optimal antibody dilution buffer.

#### 3.3.3. Antibody Amount

To obtain an optimal antibody amount level for the assay development, five levels of antibody amount, including 0.5, 1.0, 1.5, 2.0, 2.5, and 3.0 μL (17.0 mg/mL), were used to couple with TRFM. As presented in Figure 3C, the T line fluorescence intensity gradually increased with an increase of antibody amount. When the antibody amount reaches saturation, the inhibition basically remained unchanged. From the histogram analysis, the inhibition rate is basically unchanged or even a little decreased when the antibody amount is 2 μL. Considering both the sufficient performance and cost saving, the optimal antibody amount was selected as 2 μL herein (3.4 × 10^−2^ mg).

#### 3.3.4. Usage of Probe

The quantity of TRFM-Ab is directly related to the color development and cost of the test strip, the usage of TRFM-Ab was then optimized in this work. The optimization results (Figure 3D) showed that the signal intensity of the T and C lines deepened gradually with the increasing of probe volume and reached saturation when the volume was 8 μL. Thus, 8 μL of the probe was selected as the optimal usage for the further investigation.

To further improve the detection performance, ion concentration of standard diluent, methanol content in diluent, coupling pH, time of coupling reaction, and blocking time were optimized in this investigation, too (Appendix A). All optimized conditions are summarized in Table 1. The screening criteria combined the T/C value, where B and B_0_ are the ratio of T/C values with and without propiconazole in the sample solutions.

### 3.4. Sensitivity

According to the test procedure, a series of concentrations was selected to test with the strip. As presented in Figure 4A, when the propiconazole concentration is 100 ng/mL (green box), it can be seen that the T line intensity is significantly weaker than the control group (0 ng/mL). Therefore, 100 ng/mL was selected as the visual limit of detection (vLOD). Furthermore, a series of propiconazole standard solutions with different concentration were tested by FIC-Q1 fluorescence reader, and the obtained calibration curve (Figure 4B) showed a nonlinear fitting relationship between the B/B_0_ and the propiconazole concentration with a high coefficient of determination (R^2^ = 0.987). In addition, the qLOD was 1.92 ng/mL. The National Food Safety Standard Maximum Residue Limits of Pesticides in Foods (GB 2763-2019) sets maximum residue limits of propiconazole in other vegetables is 0.05 mg/kg. As shown in Table 2, the LOD of our proposed method was much lower than the national maximum residue and LODs using instrumental methods, demonstrating the applicability of this TRFMs-LFIA. Compared with the two reported propiconazole immunoassay methods [25,26], our method showed slightly higher LOD. Maybe the less sensitive polyclonal antibody we used was responsible for this phenomenon. However, the proposed TRFMs-LFIA not only fully meets the actual detection requirements, but also offers a sensitive as well as convenient immunoassay for propiconazole screening.

### 3.5. Specificity

According to the literature [47], the T/C value was chosen as the ordinate, and a histogram was used to intuitively reflect the comparison between different drugs and the blank. As shown in Figure 4C, compared with the negative control, the presence of propiconazole makes the T line disappear completely, while the other analogs did not cause obvious changes to the T line. When the buffer contains propiconazole, the T/C value of the green band is less than 0.2. The T/C value of the blue-green band with other analogs is more than twice the green band and adding other related molecules may affect the fluorescence intensity of C line, thus cause T/C value to be larger than the negative control group. This indicated that the proposed TRFMs-LFIA is highly specific for the detection of propiconazole.

The results for the 3D models (Figure 5) also confirmed the feasibility of the above results. At the lowest energy conformation, 13 triazoles and hapten AZC-HS have similar structural areas that are all in accordance with the arrangement of the 1,3-dichlorobenzene on the top, the trinitrogen ring at the bottom, and the side chain on the right. The charge distribution of hapten what we used is similar to propiconazole and the transition from benzene ring (A small amount of negative charge) to trinitrogen ring (positive charge), which implies that the obtained antibody demonstrated a good specificity. For the other 12 triazole compounds, the benzene rings of difenoconazole, diniconazole, hexaconazole, tebuconazole, paclobutrazol, flusilazole, cyproconazole, triadimenol, bitertanol standard, and triadimefon show a strong negative charge. At the same time, the positive charge of myclobutanil is mainly concentrated in the central area. Therefore, the obtained antibody cannot recognize the 12 triazole compounds, and thus demonstrated high specificity. This is also reasonable and explainable in according with the molecule modeling.

### 3.6. Recovery

As shown in Table 3, the coefficient of variation is equal to the ratio of the standard deviation to the mean and the average recovery rate of spiked samples ranged from 78.6% to 110.7% with corresponding CVs below 16%. This indicated that it could be acceptable for a rapid screening method. The good recovery can also confirm that the chiral hapten design strategy was efficient successful, since the resultant antibody can well recognize the racemate.

### 3.7. Analysis for Blind Samples

To verify the feasibility of this method in real-life samples, brassica campestris, lettuce, and romaine lettuce were chosen and tested using the TRFMs-LFIA. The vLOD of TRFMs-LFIA in samples for propiconazole was slightly higher than that in PB, which might be attributed to the effect of the complexed food matrix. As illustrated in Appendix A, with increasing propiconazole concentration, the fluorescence intensity of T line becomes weaker under UV light. When the propiconazole concentration of brassica campestris is 150 ng/mL (red box), it can be seen that the T line intensity is significantly weaker than the control group (0 ng/mL). Similarly, 200 ng/mL was selected as the visual limit of detection (vLOD) of lettuce and romaine lettuce. Therefore, the corresponding vLOD of the brassica campestris, lettuce and romaine lettuce were 150, 200, and 200 μg/kg, respectively.

Furthermore, blind samples (four samples of each type of vegetable) were detected to evaluate the suitability of the TRFMs-LFIA for in practical applications. The data of the TRFMs-LFIA were consistent with that of the HPLC-MS/MS (Table 4). Meanwhile, a satisfactory correlation for the detection of propiconazole (Y = 1.15X − 0.74, R^2^ = 0.974) was obtained with the two methods (Appendix A), indicating that the established TRFMs-LFIA had excellent reliability and could provide the point-of-care quantitative detection of propiconazole in brassica campestris, lettuce, and romaine lettuce.

To our surprise, propiconazole residues were detectable in romaine lettuce, another common vegetable on the dining table of Cantonese. It is also reported for the first time that propiconazole residue was found in romaine lettuce so far. Therefore, a comprehensive market spot check for the residue of propiconazole in romaine lettuce have to be conducted to clarify the residual status, and it is recommended that the regulatory authorities need to include the lettuce as a regulatory object.

For brassica campestris, four blind samples from the two markets all led to the detection of propiconazole residues. By calculating the dietary exposure risk entropy based on the average positive detection value (0.026 mg/kg) [11], despite the dietary exposure risk being low at present, it is necessary to monitor the propiconazole residue in vegetables, because propiconazole is not registered for use on any leafy vegetables by any country and organization.

## 4. Conclusions

In conclusion, a TRFMs-LFIA for the on-site detection of propiconazole in brassica campestris, lettuce, and romaine lettuce was developed for the first time. The corresponding vLOD for propiconazole in brassica campestris, lettuce, and romaine lettuce were 150, 200, and 200 μg/kg, respectively. Blind samples were analyzed by both the TRFMs-LFIA and HPLC-MS/MS, and a good correlation between the two methods was obtained. The application of the proposed method to blind market samples, abusing risk, was also found to exist in other agricultural products besides brassica campestris for the first time. Therefore, the established method provided an idea and rapid screening tool for propiconazole monitoring and abusing risk assessment.

## Figures and Tables

**Figure 1 foods-11-00324-f001:**
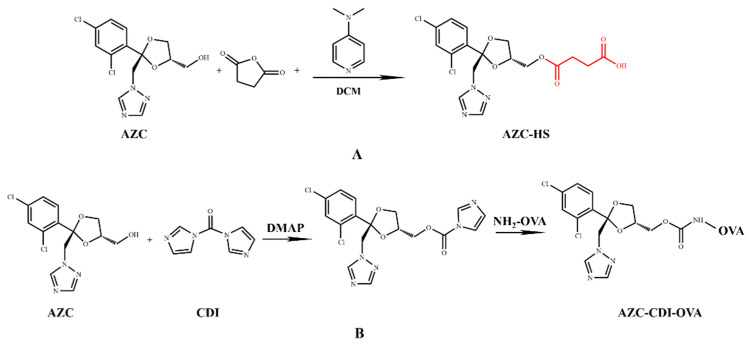
(**A**) Synthesis of the AZC-HS hapten. (**B**) Hapten-carrier conjugation of AZC-CDI-OVA.

**Figure 2 foods-11-00324-f002:**
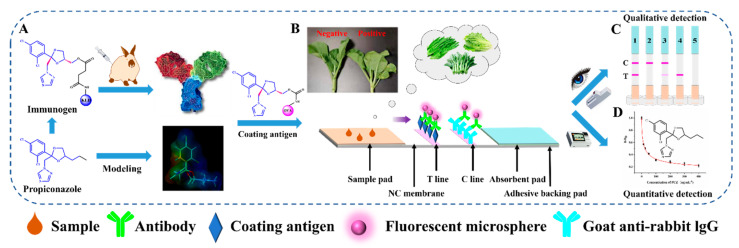
(**A**) Design of hapten and molecular modeling. (**B**) Schematic diagram of TRFMs-LFIA for propiconazole in brassica campestris, lettuce and romaine lettuce. (**C**) The test results from the LFIA. 1, Negative result; 2,3, positive result; 4,5, invalid results. (**D**) Quantitative detection.

**Figure 3 foods-11-00324-f003:**
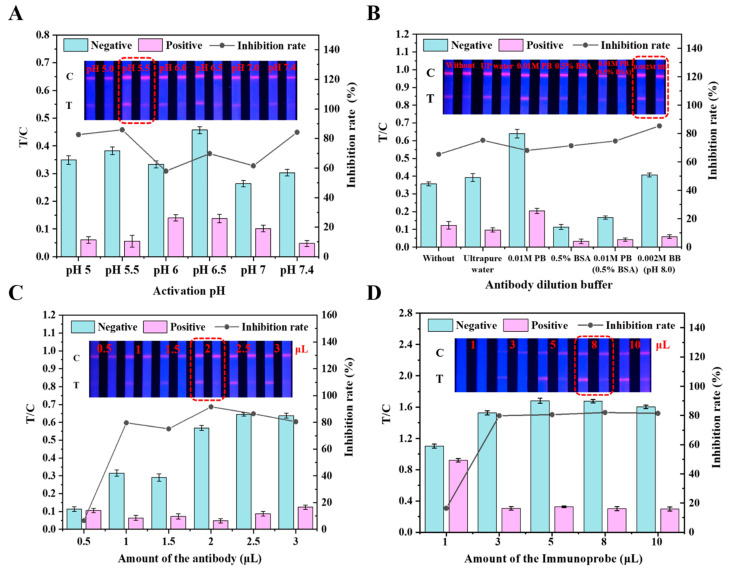
Optimization of working conditions. All the optimize conditions were evaluated by the negative (0 ng/mL) and positive (100 ng/mL). The values of T/C below were calculated from the pictures above by FIC-Q1 fluorescence reader and the screening criteria combined the T/C value, where B and B_0_ were the ratio of T/C values with and without propiconazole in the sample solutions. Inhibition rate is equal to 1-B_0_/B. (**A**) Activation pH, (**B**) Antibody dilution buffer, (**C**) Antibody amount, (**D**) Immunoprobe amount.

**Figure 4 foods-11-00324-f004:**
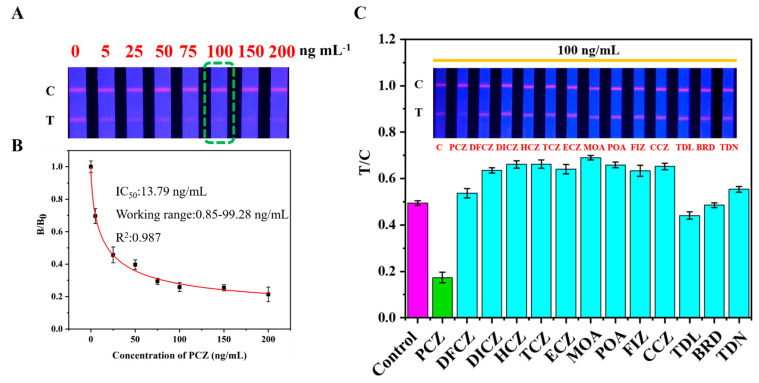
Assay performance of TRFMs-LFIA for propiconazole detection. (**A**) Detection results for propiconazole standard of different concentrations by the TRFMs-LFIA. Green rectangular box represents the vLOD concentrations of propiconazole by the TRFMs-LFIA. (**B**) Calibration curve of propiconazole in standard buffer. (**C**) Specificity of the TRFMs-LFIA for propiconazole detection. The concentrations of propiconazole (PCZ), difenoconazole (DFCZ), diniconazole (DICZ), hexaconazole (HCZ), tebuconazole (TCZ), epoxiconazole (ECZ), myclobutanil (MOA), paclobutrazol (POA), flusilazole (FIZ), cyproconazole (CCZ), triadimenol (TDL), bitertanol standard (BRD), and triadimefon (TDN) are all 100 ng/mL.

**Figure 5 foods-11-00324-f005:**
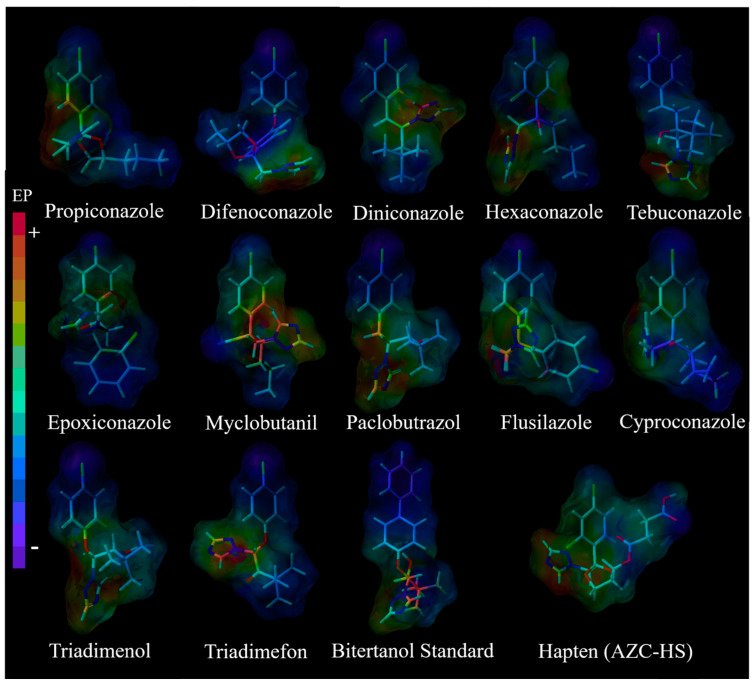
Lowest energy conformations and molecular electrostatic potential isosurfaces of 13 triazole compounds and hapten (Blue region represents negative charge, red region represents for positive charge. The darker the color, the stronger the charge).

**Table 1 foods-11-00324-t001:** Working conditions of the TRFMs-LFIA.

Working Conditions	Optimal Value
Ion concentration of standard diluent	0.2 M PB
Methanol content in diluent	10%
Activation pH	0.05 M MES (pH 5.5)
Coupling pH	0.02 M BB (pH 8.0)
Antibody dilution buffer	0.002 M BB (pH 8.0)
Antibody amount	3.4 × 10^−2^ mg (per strip)
Time of coupling reaction	45 min
Blocking time	60 min
Immunoprobe amount	8 μL

**Table 2 foods-11-00324-t002:** Comparison of performance of different methods.

Method	Matrix	The Detection Lim (itmg/kg)	References
GC-MS	Banana	0.02	[19]
LC-MS/MS	Pepper soil	0.005 0.0015	[22]
LC-MS/MS	Soil	0.004	[21]
SPE-GC-μECD	Vegetable	0.01	[45]
GC-ECD	Groundwater	2	[15]
GC-MS	Snow peas	0.003	[20]
HPLC-MS	Soil	0.005	[46]
GC	Wolfberry	0.006	[16]
GC-ICA	Vegetable	0.00013	[25]
Ic ELISA	Vegetable	0.00026	[26]
TRFMs-LFIA	Brassica campestris	0.00192	This work

**Table 3 foods-11-00324-t003:** Recovery of propiconazole in brassica campestris, lettuce and romaine lettuce samples detected by TRFMs-LFIA (*n* = 3).

Samples	Spiked Level	Found ± SD	Recovery	CV
	(ng/g)	(ng/g)	(%)	(%)
brassica campestris	10.0	8.6 ± 0.8	86.3	9.1
	40.0	44.3 ± 5.4	110.7	12.2
	80.0	78.6 ± 12.3	98.3	15.6
lettuce	10.0	8.7 ± 0.5	86.6	5.6
	40.0	36.7 ± 5.1	91.6	13.9
	80.0	83.2 ± 8.3	104.0	9.9
romaine lettuce	10.0	8.1 ± 0.7	81.2	8.8
	40.0	31.5 ± 3.7	78.6	11.8
	80.0	71.1 ± 9.7	88.9	13.6

SD, Standard deviation. CV, Coefficient of Variation.

**Table 4 foods-11-00324-t004:** Comparison of propiconazole using TRFMs-LFIA and HPLC-MS/MS in blind samples (brassica campestris, lettuce and romaine lettuce) (*n* = 3).

Assay		TRFMs-LFIA		HPLC-MS/MS	
Samples	Number	Test Value (Mean ± SD, ng/g)	CV (%)	Test Value (Mean ± SD, ng/g)	CV (%)
brassica campestris	Sample 1	14.2 ± 1.6	11.5	19.0 ± 1.8	9.4
	Sample 2	64.2 ± 5.3	8.2	71.3 ± 3.5	4.9
	Sample 3	13.5 ± 1.8	13.4	9.5 ± 0.8	8.9
	Sample 4	6.6 ± 0.9	13.8	4.6 ± 0.3	7.4
lettuce	Sample 5	ND	-	ND	-
	Sample 6	ND	-	ND	-
	Sample 7	ND	-	ND	-
	Sample 8	ND	-	ND	-
romaine lettuce	Sample 9	ND	-	ND	-
	Sample 10	ND	-	ND	-
	Sample 11	ND	-	ND	-
	Sample 12	8.8 ± 0.8	9.6	12.4 ± 1.3	10.8

ND, Not detected. -, unavailable.

## Data Availability

Data is contained within the article (or Appendix A).

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
