# Peer review of "Antibody Generation and Rapid Immunochromatography Using Time-Resolved Fluorescence Microspheres for Propiconazole: Fungicide Abused as Growth Regulator in Vegetable"

_foods, 2022, doi:10.3390/foods11030324_

Round 1

Reviewer 1 Report

The article titled “Antibody generation and rapid immunochromatography using time-resolved fluorescence microspheres for propiconazole: fungicide abused as growth regulator in vegetable” developed both items as the antibody production and immunoassay for propiconazole. Although this work could be interesting for the readers of the Journal, in my opinion, some important parts have to be rewritten to fulfil the requirements for its publication.

Next, I will indicate some questions or comments.

  1. In page 4, paragraph 4, probably there is a typographic error: “The distance between the T and C line was 8 nm”.
  2. The spiking of vegetables is not clearly specified. Were the fungicide directly added to the leaves and then performed the extraction procedure?
  3. In epigraph 2.10, principles of TRFM-LFIA, there are lot of misunderstandings. In my opinion, this epigraph has to be rewritten.

For instant, in the first paragraph, authors indicate that even with or without the presence of the analyte two fluorescent bands will appear. However, in the third paragraph, they say that when analyte is present only one band will appear, which is more consistent with the format of the assay.

In addition, the second paragraph is not clear at all. What the micropore are? The performance of the assay has to be clearly exposed. How and where are the sample and the TRFM reagent put in contact for reaction and so on?

It is said that T/To was used to quantify the result, where T and To represent fluo-rescence intensity at the detection line of the negative and positive analytes, respectively. However, later, authors talks just about the relation T/C.

  1. In epigraph 2.11, authors said that standard curves were plotted based on the B/Bo (T/C) against the logarithmic of the propiconazole concentration, but it is not the case of figure 3B. Also in page 9, authors claims that the obtained calibration curve (Figure 3B) showed an inverse linear relationship between the B/B0 and the propiconazole. This is not congruent with the figure. In this figure is not possible to visualize the working range and limit of quantification.
  2. Also in epigraph 2.11, the evaluation of specificity is included, not the accuracy and precision, which are in 2.12. All of them are the “Performance of the TRFMs-LFIA.
  3. In page 7, first paragraph, there is a confusion about what protein is used for immunogen or coating antigen (BSA, OVA, KLH). It has to be clarified in the text.
  4. Epigraph 3.3 is also very confusing. In my opinion it has to be differentiated the optimization steps to produce the reagent TRFM-Ab (pH, dilution buffer, Ab amount…) related to the optimization of the assay (TRFM-Ab amount, times, etc.).

By other hand, there are no coincidences between the letters of figure S5 used in the text and the presented histograms in figure. In addition, there is not concordance between the 8 uL of the quantity of probe selected in the text for the TRFM-Ab optimization ant the figure S5.

Finally, in the epigraph 3.3.4 other optimizations are commented in a mixed way and referred to the table 1. Are they less important than the previous ones?

  1. Figure 3 is put before figure 2.
  2. I do not understand how the vLOD is calculated. In figure 3A, there are other concentrations with differences in T line related to the zero concentration. It looks like that this value (100 ng/mL) could be the maximum amount of analyte to see with naked eyes. Furthermore, this value is higher than the high value in the working range (99.028 ng/mL)
  3. How is calculated the qLOD od 1.92 ng/mL?
  4. In the sentences of page 9 related to the other analytical methods, are these values of concentration the LODs of the techniques for this analyte? It is not clear what authors want to say. References 15 to 22 are not taken in consideration for comparison. A more detailed comparison of methods will be valuable.
  5. In the text of figure 3, the sentence “Green rectangular box represents the vLOD concentrations of propiconazole by the TRFMs-LFIA” correspond to (A).
  6. In figure 3C, some comment has to be added explaining why the bars for other related molecules are quite higher than the negative control.
  7. In table 2, a line between different vegetables will be helpful to improve the quality of the table.

In table 3, there are not concentration units. Perhaps it is not sense to differentiate the lettuce samples, since there are not detected.

Reviewer 2 Report

Please see attached Review Report.

Reviewer 3 Report

The manuscript “Antibody generation and rapid immunochromatography using time-resolved fluorescence microspheres for propiconazole: fungicide abused as growth regulator in vegetable” presents a new immunoanalytical method for the fungicide propiconazole. Propiconazole seems to be hazardous to human health; thus, a new, easy-to-perform analytical technique for determining this fungicide in food samples is of interest for a wide range of readers. Moreover, the chemical approaches the authors have followed to prepare the immunogen and the coating antigen used are also interesting for those working in the development of hapten-immunochemical assays.   Nevertheless, there are several weak points and/or obscure issues, which should be suitably addressed.

Introduction: The authors should present any previously published immunoanalytical methods for propiconazole. If the immunochromatography they have developed is the first immunoanalytical method for propiconazole, they should clearly declare it.

2.5. Antibody generation, p. 4, line 1: “New Zealand white rabbits (6-7 weeks age) were immunized…”. The Ethical Review document (Figure S1) refers to a rather large number of rabbits (10 female plus 10 male). Have all these rabbits been immunized? Have all immunized rabbits developed anti-propiconazole antibodies of desirable immunochemical characteristics? (Note: The Ethical Review document also refers to “mice”; is there any specific reason why?)

2.8. Fabrication of the lateral flow strip, p. 4, line 6 of 2.8: “The distance between the T and C line was 8 nm”. Please, verify.

2.9. Sample preparation: Please, add a short description of the preparation of spiked samples

2.9. Sample preparation, p. 5, lines 1-3: “…reconstituted with 2 mL of PB (200 mM, pH 7.4) containing 10% (v/v) methanol and filtered through 0.22 μm filter membrane, and then diluted four times with PB (200 mM, pH 7.4) containing 10% (v/v) methanol for TRFMs-LFIA analysis”. Please, provide specific final volume (corresponding to 2 g of vegetables)

2.10. Principle of TRFMs-LFIA: In general, 2.10 needs to be rewritten, in a clearer way, e.g.: 2.10. Principle of TRFMs-LFIA, p. 5, lines 10-12 of the first paragraph of 2.10: “Thus, two fluorescent bands appear under UV light due to the accumulation of TRFM-Ab conjugates…” Do you mean “Thus, in case of negative samples two fluorescent bands appear under UV light due to the accumulation of TRFM-Ab conjugates…”?

2.10. Principle of TRFMs-LFIA, p.5, line 4 of the second paragraph of 2.10: “…were vertically inserted into the micropores…”. Could it be possible to show schematically the “micropores” in Figure 1?

3.1. Hapten design and antibody evaluation, p. 7, first paragraph, lines 2-3: “…and the hapten AZC was directly coupled to the OVA to obtain the coating antigen”. Please, add a Scheme showing preparation of AZC-DPI-OVA (along with that showing synthesis of AZC-HS) in Figure S2, for better clarity. Figure S2 might be removed from Supporting Information to the main manuscript.

3.3.1. Activated pH (Activation pH??), p. 7, line 6 of 3.3.1: “…(Figure S5A)…” Do you mean “…(Figure S5C)…”??

3.3.1. Activated pH (Activation pH??), p. 7, lines 6-7 of 3.3.1: “When the pH was at 5.5, the fluorescence intensity…”. Do you mean “When the pH during preparation of the TRFM labeled antibody was at 5.5, the fluorescence intensity…”?

3.3.2. Antibody dilution buffer, p. 8, line 2: “…(Figure S5B)…” Do you mean “…(Figure S5E)…”??

3.3.3. Antibody amount, p. 8, line 3 of 3.3.3: “…(Figure S5C)…” Do you mean “…(Figure S5F)…”??

3.3.4. Usage of probe, p. 8, line 3 of 3.3.4: “…(Figure S5D)…” Do you mean “…(Figure S5I)…”??

3.4. Sensitivity, p. 9: In order to avoid confusion, please, explain explicitly the approach followed to directly compare the qLOD of the method developed (expressed in ng/mL) with the MRLs set for propiconazole (expressed in mg/kg)

3.5. Specificity, p. 10: Please, explain the criteria used to select the putative cross-reacting triazole compounds. In addition to structure similarity issues, has any of them been found in vegetables?

3.6. Recovery, p. 10: Please, describe how exactly the spiked samples have been prepared (see also the comment for 2.9, Sample preparation) and explain correlation between ng/mL and ng/g

Table 2: There are no footnotes explaining “a” and “b”

Table 3: Please, add units (column 3, column 5)

Figure S4, legend, line 2: “…Lane 1, Before purification. Lane 2, the reduction results…”. It has to change into “…Lane 1, Before purification. Lane 2, After purification…”

Round 2

Reviewer 1 Report

Authors have corrected the questions asked. In my opinion, the article can be published in the revised version.

Author Response

Thanks. 

Reviewer 3 Report

In the revised version of Manuscript Foods-1486116, the authors have addressed most of the issues pointed out during the reviewing process. In my opinion, the manuscript can be accepted for publication, after minor  revision.

Comments on the revised manuscript

  1. Page 2, end of third paragraph, “However, immunochromatography has become the most important part of the rapid detection filed …”: Please, omit the word “However”. Also, please, specify whether any other immunoassays/immunochromatography methods have been published so far.
  2. Page 3, 2.3, “Hapten synthesis”: this may change in “Synthesis of hapten AZC-HS”, for better clarity.
  3. Page 5, 2.9, Sample preparation: According to the authors’ clarification, 2 g of vegetable samples correspond, in the first place, to 2 mL of sample volume (which is further 4-fold diluted, to a final volume of 8 mL). In this context, it has been clear how ng/mL can be translated directly into ng/g.
  4. Page 7, end of second paragraph, “And the conjugate curves (Figure S2A and B) showed the conjugate has the characteristic peaks of both the hapten at 320 nm and the carrier protein at 280 nm, indicating successful conjugation between the hapten and the carrier (BSA or OVA)…”. However, in the UV-Vis spectrum of the hapten AZC-HS, no peak seems to appear at 320 nm (Figure S2A). Is this so? Could you, please, correct me/explain?
  5. Page 8, 3.3.2, Activation pH, last words of the paragraph: “…as the optimal activated pH”: this should change to “…as the optimal activation pH”
  6. Page 8, 3.3.4., last paragraph: “…were optimized in this investigation, too (Figure S5E and I)”. This should change into: ““…were optimized in this investigation, too (Figure S4, A-E)”.
  7. Page 9, Figure 3A, x-axis: “The activated pH of buffer”: it may be better to change to: “Activation pH (MES solution)”.
